# Medical staff's perspectives on patients' anxieties and interventions in a rehabilitation ward: A qualitative study

**Taiki Yoshida**[1], **Yoshitaka Wada**[2], **Shintaro Uehara**[1], **Asuka Hirano**[3],
**Kazuki Ushizawa**[2], **Hirofumi Maeda**[2], **Daisuke Matsuura**[2], **Yohei Otaka**[2*]

**1** Faculty of Rehabilitation, Fujita Health University School of Health Sciences, Aichi, Japan, **2** Department of Rehabilitation Medicine, Fujita Health University School of Medicine, Aichi, Japan, **3** Department of Rehabilitation, Fujita Health University Hospital, Aichi, Japan

\* otaka119@mac.com

## Abstract

### Background

Anxiety and depression in rehabilitation patients can adversely impact clinical outcomes. They may have anxieties about the differences in their physical conditions and living environments compared to before hospitalization. Although medical staff address patients' anxieties, the content of anxieties and the type of intervention have not been clarified. This study aimed to highlight the contents of anxieties and interventions for patients undergoing rehabilitation, based on medical staff's perspectives.

### Methods

Seventeen medical staff were interviewed about the anxieties they perceived patients experience at the convalescent rehabilitation ward (early, middle, and late phases of hospitalization) and the corresponding interventions. Text mining and hierarchical cluster analysis were used to classify the contents of anxieties and interventions. This study was conducted based on the consolidated criteria for reporting qualitative research.

### Results

Patients' anxieties were classified into six clusters. Among the clusters, prospects for rehabilitation plans, hospital life (e.g., unfamiliar hospital environment), and family situation (e.g., concerns about family life at home) were identified in the early to middle phases, and life at home after discharge was identified in the late phase. The prognosis of physical function and prospects of social life (e.g., return to work) were identified throughout all phases. The types of interventions for these anxieties were classified into eight clusters. The medical staff provided information about patients'

**Data availability statement:** All relevant data are within the paper and its Supporting information files.

**Funding:** This study was supported by JSPS KAKENHI Grant-in-Aid for Early-Career Scientists (Grant no. 22K17579). The funders had no role in study design, data collection and analysis, decision to publish, or preparation of the manuscript.

**Competing interests:** The authors have declared that no competing interests exist.

prospects and helped them contact family members in the early phase. In the middle phase, feedback on patients' improvement in physical function was incorporated. In the late phase, information on social resources was provided to address anxieties about life after discharge.

## Conclusion

This study showed that patients' anxiety and interventions varied according to hospitalization phases. The findings underscore appropriate ways and the timing of interventions to keep patients in a better psychological state, potentially leading to better rehabilitation outcomes.

## Introduction

Disease onset is often a sudden event for many patients, and some patients require hospitalization. Hospitalization is often a stressful experience that can include negative stress for patients. In recent years, increasing attention has been paid to the psychological states of patients throughout all phases, from disease onset to the chronic phase [1–3]. The negative stress leads to problems affecting psychological states and eventually results in depression and anxiety disorders [4]. Notably, patients with stroke show a higher prevalence of depression and anxiety than the general population [5,6], which is known to negatively affect rehabilitation outcomes. For example, depression negatively affects rehabilitation outcomes, such as independence in activities of daily living [7–9], functional improvements [10], the home discharge rate [7], the length of hospital stay [7,11], and quality of life [12]. Likewise, anxiety disorders negatively impact quality of life [13] and the motivation for rehabilitation [14]. Therefore, it is important to provide interventions for negative psychological states before the onset of depression and anxiety disorders.

Regarding interventions for negative psychological states, psychotherapy is commonly implemented [15–18]. Several studies have examined structured interventions aimed at addressing negative psychological states for patients. For example, goal-setting interventions by multidisciplinary medical teams have been shown to improve functional outcomes and mood in stroke patients [19]. This approach highlights the importance of personalized care, as goals are adapted to each patient's condition and recovery processes. Similarly, person-centered care strategies, which have their origins in dementia care [20], have recently attracted attention as psychological intervention approaches for patients undergoing rehabilitation, emphasizing phase-sensitive communication and support adapted to the patients' current psychological states [21]. Importantly, cultural contexts can influence both patients' psychological states and the interventions conducted by medical staff. For example, in Japan, cultural norms that emphasize close interpersonal relationships and the reconstruction of self-identity through social relationships may influence the recovery of individual psychological states, rather than through individual autonomy or

separation from social expectations [22]. These cultural contexts highlight the importance of examining psychosocial care in specific sociocultural settings.

In clinical settings, psychotherapy is mainly conducted by qualified clinical psychologists. However, medical staff who are non-psychologists or do not have a psychology background may also work to address patients' negative psychological states using interactions incorporating elements of psychotherapy. Indeed, it has been suggested that even non-psychologist medical staff, such as physical and occupational therapists, could effectively reduce negative psychological stress in patients [23,24]. Although specialized interventions by clinical psychologists are usually scheduled for a designated session, non-psychologist medical staff appear to deal with patients in their respective daily duties rather than having a session solely for this purpose. In many cases of psychotherapy sessions by clinical psychologists, they assess the patients' psychological anxieties and provide interventions that best fit the specific symptoms. Even for non-psychologist medical staff, the same processes—assessments first and then interventions—are needed, but the difficulty is that they have to execute them in the limited time of daily duties. Therefore, it would be beneficial for them to know the contents of patients' anxieties and related interventions in advance, to provide efficient treatments.

In clinical settings, the patients' psychological states, for example, symptoms of depression and anxiety disorders, are assessed using specific evaluation scales, including the Self-rating Depression Scale [25], Geriatric Depression Scale [26], Hamilton Depression Rating Scale [27], Hospital Anxiety and Depression Scale [28], and Beck Anxiety Inventory [29]. Although these evaluation scales can assess the presence of depression and anxiety disorders, they do not identify specific contents of anxieties that lead to negative psychological states. Furthermore, patients' anxieties may change over time during hospitalization. In particular, patients undergoing intensive rehabilitation can improve their physical function [30,31] and their anxieties may change accordingly. However, specific contents of patients' anxieties and their temporal changes, alongside the interventions to address patients' psychological states, remain unclear.

This study aimed to provide an overview of the contents of anxieties potentially negatively affecting patients' psychological states and their changes over time and identify corresponding interventions for subacute patients undergoing intensive rehabilitation in the hospital. Subjective data based on the patients' own experiences collected through interviews with patients may include recall bias [32] and recency effects [33]. Additionally, patients might only mention contents related to their own diseases. Conversely, since medical staff routinely observe patients throughout all phases of hospitalization, they can offer insights that are not limited to a specific disease. Therefore, interviews with medical staff are suitable for efficiently collecting data about patients with various diseases and at different phases of hospitalization. While the analysis of qualitative data such as interviews can be influenced by biases from the analysts' interpretations, text-mining methods can be employed to reduce such biases [34]. Therefore, we used a qualitative approach (i.e., semi-structured interviews) [26] and text-mining analysis based on the medical staff's perspectives to examine the contents of patients' anxieties and corresponding interventions.

## Materials and methods

### Study design

This study was conducted based on the consolidated criteria for reporting qualitative research (COREQ) [35] and in accordance with the Declaration of Helsinki. It was approved by the Ethics Review Committee of Fujita Health University (approval number HM22-373). In the study, we interviewed medical staff regarding patients admitted to the convalescent rehabilitation ward at Fujita Health University Hospital. The convalescent rehabilitation wards are systems for subacute rehabilitation and their services are covered by government medical insurance in Japan [30]. In many cases, patients are usually admitted to the acute care hospital after disease onset and receive treatment for approximately 1–2 months. Subsequently, when medical staff think patients require additional rehabilitation treatment, they are transferred to the convalescent rehabilitation ward, except for mild cases wherein they can independently perform daily activities at home. The patients receive intensive rehabilitation treatment for an average of 2–3 months. More than half of the patients who

are admitted to the convalescent rehabilitation ward at Fujita Health University Hospital are those who have experienced stroke, and the rest have experienced traumatic brain injury, fractures, or spinal cord injury.

We conducted an interview of medical staff working in a convalescent rehabilitation ward and analyzed transcripts generated from the audio data using text mining techniques. Semi-structured interviews were conducted with guiding questions, including "What are the anxieties in patients during the early/ middle/ late phase of hospitalization in a convalescent rehabilitation ward" and "How do you address those anxieties?" Interviewees were asked to talk about their current and past experiences with patients in the convalescent rehabilitation ward. The early phase of hospitalization was defined as the period of approximately two weeks after admission, until the completion of medical staff conferences to share a patient's rehabilitation goal. The late phase of hospitalization was defined as the period of approximately two weeks prior to discharge when the date and place of discharge were determined. The middle phase was defined as the period between the early and late phases of hospitalization.

The interviews were carried out by either TY, an occupational therapist (OT), or YW, a physiatrist at the Fujita Health University Hospital; they had 13 and 9 years of clinical experience, respectively, at the time the study was conducted. TY had substantial experience in qualitative research. TY and YW discussed the interview procedures before conducting the interviews. To ensure consistency and reliability, an interview guide and a set of prompts were prepared and used during the interviews (Supplemental Data). Although the interviewers and participants worked at the same hospital, they had no personal or financial relationships with each other. The participants understood that the researchers were members of a psychosocial support group for patients in the same ward. Interviews were conducted on a one-on-one basis in a quiet room or space of the hospital between February and June 2023. All interview content was recorded using an integrated circuit recorder (SR502J, iFLYTEK JAPAN AI SOLUTIONS Co., Ltd.) with pre-obtained permission from the participants. The interview topic guide was developed based on discussions between two OTs (TY and KU), a physical therapist (PT) (AH), and two physiatrists (YW and YO). KU was an occupational therapist with 8 years of clinical experience and had experience in qualitative research. AH was a physical therapist at the same hospital with 18 years of clinical experience. YO was a physiatrist at the same hospital with 28 years of clinical experience. Explanations for each question were prepared and shared among the interviewers; the meaning of the question was explained using these shared understandings before each question was asked. Given the psychological burden on the participants, they were informed that the interview could be stopped immediately if any such burden was felt.

## Participants

We recruited 17 medical staff (mean [standard deviation] years of clinical experience, 6.6 [3.2]) including two physiatrists, three PTs, five OTs, four speech-language pathologists (SLPs), two nurses, and one certified care worker working in the Fujita Health University Hospital through convenience sampling. The inclusion criteria involved participants having worked in the convalescent rehabilitation ward for at least one year. The participants who did not provide informed consent were excluded from this study. To recruit participants, we displayed posters explaining the purpose and procedures of the study to the ward. Individuals who expressed interest were recruited to participate in this study. Each participant was interviewed only once. All participants provided written and verbal informed consent before participating in the study. The interview was conducted after explaining that the interview contents would not interfere with their work performance evaluation and with the approval of the participants. To justify the sample size and inclusion process, we referenced previous studies indicating that theoretical saturation in qualitative interviews is typically achieved with approximately 12 participants [36]. Therefore, we aimed to recruit beyond this number while monitoring saturation. Then, after conducting more interviews than the number of participants in the previous study, we determined that theoretical saturation was reached when no new themes emerged from three consecutive interviews, regardless of the participants' professions. We concluded that further sampling was unnecessary at that point and ceased recruitment accordingly.

## Analyses

The recorded interview data were transcribed into text, which was then thoroughly read to extract meaning from each data source. To ensure objectivity, transcription was conducted by an independent third party. To preserve anonymity, personally identifiable data were not included in the analysis. We performed the text mining technique and hierarchical cluster analysis (HCA) [37] with the transcript data using KH Coder [38,39]. We performed text mining by compiling all the text data concerning patients' anxieties negatively affecting their psychological states during the entire phase of hospitalization and the early, middle, and late phases, respectively. We only used nouns in the HCA, and we included nouns with a frequency of at least 10 occurrences in the entire duration of hospitalization and at least five occurrences in the early, middle, and late phases of the hospitalization. To cleanse the data before text mining, we corrected spelling and punctuation errors. We extracted nouns as target words and then unified synonymous words into a single word while preserving the meaning of the text. This process was based on a list of extracted words and references from the original response text. We used Ward's method [40] to measure co-occurrences. The nouns were classified into clusters based on the Jaccard distance and we located the resultant HCA dendrogram using KH Coder [38,39]. We determined the threshold of the agglomeration dissimilarity coefficient to easily interpret and label each cluster. English translations written using Japanese nouns may not necessarily contain one-word nouns. For the analysis process, TY, AH, and KU labelled each cluster. To ensure that the labelling accurately reflected the participants' intended meaning, they reviewed the words included in each cluster identified by KH Coder and the original transcripts. Furthermore, we matched the contents of patients' anxieties to interventions after consensus by the authors, referring to the content of the clusters and the interview data. We did not return interview transcripts or cluster results to participants for review (i.e., member checking).

## Results

The average interview time for the 17 participants was 27 min 40 s [434 s]. Interviewers A and B conducted 15 and two interviews, respectively.

### Patients' anxieties

In the interviews, from what the participants talked about, a total of 795 sentences were related to patients' anxieties. In total, 21,609 words consisting of 1,483 morphemes were identified. Among these, 477 types of nouns occurred 1,785 times in total (Table 1). These nouns were classified into six clusters based on the threshold of agglomeration dissimilarity coefficient in the cluster dendrogram, determined through discussion among the authors (Fig 1). We named these clusters as follows: 1) life at home after discharge, 2) prognosis of physical function, 3) prospects of social life, 4) hospital life, 5) family situation, and 6) prospects for rehabilitation plans (Table 2). The details for each phase of hospitalization are shown in Tables 1 and 2, and S1–S3 Figs.

**Table 1. Analyzed text information.**

| | Patients' anxieties | | | | Types of interventions | | | |
|---|---|---|---|---|---|---|---|---|
| | Entire duration | Early phase | Middle phase | Late phase | Entire duration | Early phase | Middle phase | Late phase |
| Sentences | 795 | 227 | 244 | 324 | 781 | 205 | 296 | 280 |
| Words | 21,609 | 6,531 | 6,616 | 8,462 | 25,008 | 6,922 | 9,258 | 8,828 |
| Morphemes | 1,483 | 778 | 797 | 884 | 1,705 | 884 | 1,004 | 971 |
| Types of nouns | 477 | 210 | 223 | 258 | 615 | 292 | 320 | 306 |
| Occurrences | 1,785 | 536 | 559 | 690 | 2,398 | 694 | 826 | 878 |

In sum, among the contents of anxieties, prospects for rehabilitation plans, hospital life, and family situation were identified in the early to middle phase, and life at home after discharge was identified in the late phase. The prognosis of physical function and the prospects of social life were identified throughout the hospitalization phase.

## Types of interventions

In the interviews, from what the participants shared, a total of 781 sentences were concerning interventions used during the entire hospitalization period. In total, 25,008 words consisting of 1,705 morphemes were identified. Among these, 615 types of nouns occurred 2,398 times in total (Table 1). These nouns were classified into eight clusters based on the threshold of agglomeration dissimilarity coefficient in the cluster dendrogram, determined through discussion among the authors (Fig 2). We also named these clusters as follows: 1) providing information about care services, 2) practicing movements needed for life after discharge, 3) clarifying the rehabilitation content and plan, 4) support for family and patient interactions, 5) setting and sharing goals, 6) providing practice in increasing the independent level of activities in the hospital, 7) providing information on return to work and support for life after discharge, and 8) motivating patients by providing feedback on improvements (Table 3). The details for each phase of hospitalization are shown in Tables 1 and 3, and S4–S6 Figs.

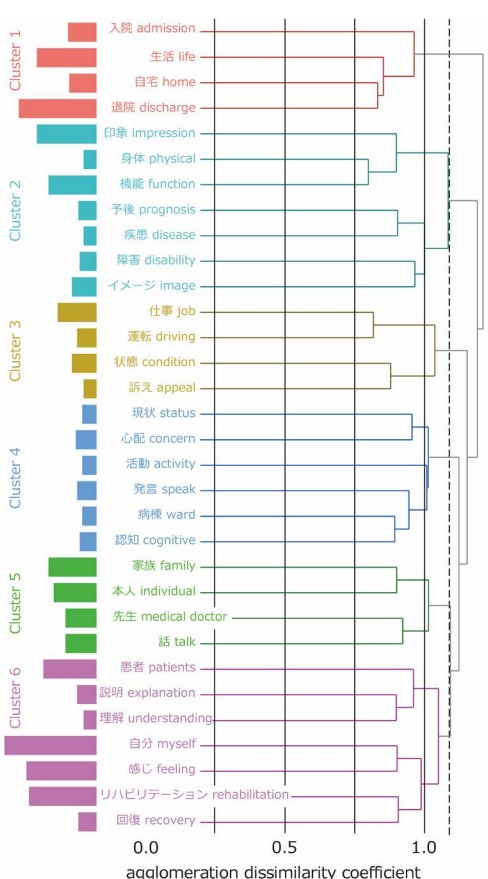

**Fig 1. Cluster dendrogram of the patients' anxieties during the entire duration of hospitalization.** Cluster 1, life at home after discharge; Cluster 2, prognosis of physical function; Cluster 3, prospects of social life; Cluster 4, hospital life; Cluster 5, family situation; and Cluster 6, prospects for rehabilitation plans. Dotted vertical line: Threshold of the agglomeration dissimilarity coefficient.

**Table 2. Clusters for patients' anxieties during the entire duration, early, middle, and late phases of hospitalization.**

| Entire duration of hospitalization | Early phase | Middle phase | Late phase | Participant narratives |
|---|---|---|---|---|
| Prospects for rehabilitation plans | Prospects for rehabilitation plans | Prospects for rehabilitation plans | | "Patients are often anxious about whether their paralysis will improve." (Nurse 1) |
| | | Lack of feelings of improvement | | "Patients tend to feel uneasy when they sense that their recovery is taking longer than expected." (OT 3) |
| Prognosis of physical function | Prognosis of physical function | Prognosis of physical function | | "Patients worry about their overall prognosis, wondering whether they'll make a full recovery or, if not, how much function they'll be able to regain." (Physiatrist 2) |
| | Prognosis of cognitive disorders | | | "Patients with aphasia or cognitive disorders often become anxious once they begin to recognize their symptoms and wonder how much they can improve." (SLP 2) |
| | | | Decline in physical function after discharge | "Patients are worried about going home, expressing fear that their condition might get worse after discharge." (OT 2) |
| Family situation | Family situation | | | "Patients who used to care for their family members often say things like, 'I need to get back soon, otherwise my family won't manage without me.'" (SLP 3) |
| Hospital life | Differences from the acute care hospital | | | "During their acute care stay, patients are often told that rehabilitation will be tough in the next phase, which makes some of them anxious about whether they can keep up with the program." (OT 3) |
| Prospects of social life | Prospects of social life | Prospects of social life | | "Patients who were managing a household or working often seem more worried about how things are going in their absence than about whether they can return. They're concerned about their family or workplace." (Physiatrist 1) |
| | | | Role reacquisition (return to work, household chores) | "Many patients feel anxious about whether they'll be able to live independently at home after discharge. Those planning to return to work often feel unsure whether they'll really be able to do their job again." (PT 3) |
| Life at home after discharge | | | Lack of information about the medical follow-up system after discharge | "Patients often ask about the type of rehabilitation they will receive after discharge, including long-term care services and outpatient therapy." (PT 2) |
| | | | Differences in the physical and interpersonal environments between the hospital and home | "Patients express concerns about living in an environment that is different from the hospital, and whether they'll be able to manage on their own." (Physiatrist 2) |

The interventions appeared to vary according to the changes in patients' anxieties during hospitalization (Table 4, S1–S3 Tables). In the early phase of hospitalization, the medical staff are more likely to provide information about patients' prospects and help them through supportive interactions, such as by providing opportunities to contact family members. In the middle phase of hospitalization, feedback on patients' improvement in physical function and ability is added to the interventions in the early phase. In the late phase of hospitalization, information on social resources, such as outpatient and nursing care services and self-exercise, is provided to address anxieties about life after discharge.

## Discussion

By interviewing medical staff, this study provides an overview of the contents of patients' anxieties among those admitted to a convalescent rehabilitation ward and the interventions for them. Through text mining and HCA, a total of six clusters for patients' anxieties and eight clusters for interventions were identified. We also found that these anxieties and interventions may change depending on the hospitalization phases. In the early phase, a lack of information about the

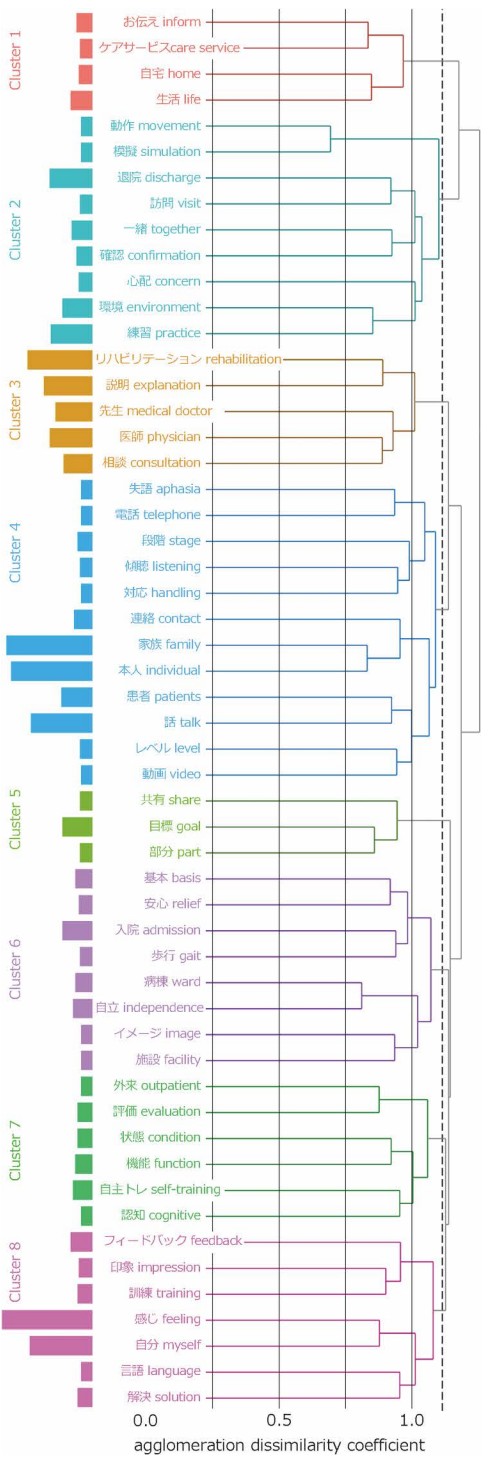

**Fig 2. Cluster dendrogram of the interventions for patients' anxieties in the entire duration of hospitalization.** Cluster 1, providing information about care services; Cluster 2, practicing movements needed for life after discharge; Cluster 3, clarifying the rehabilitation content and plan; Cluster 4, support for family and patient interactions; Cluster 5, setting and sharing goals; Cluster 6, providing practice in increasing the independent level of activities in the hospital; Cluster 7, providing information on return to work and support for life after discharge; and Cluster 8, motivating patients by providing feedback on improvements. Dotted vertical line: Threshold of the agglomeration dissimilarity coefficient.

**Table 3. Clusters for the types of interventions during the entire duration, early, middle, and late phases of hospitalization.**

| Entire duration of hospitalization | Early phase | Middle phase | Late phase | Participant narratives |
|---|---|---|---|---|
| Clarifying the rehabilitation content and plan | Explanation of the current situation by medical staff | | | "I explain the patient's current condition, including the goals we're working toward and the type of rehabilitation they're receiving." (SLP 1) |
| | Explanation of the rehabilitation treatment plan by physician | Explanation of the rehabilitation treatment plan by physician | | "It's probably best if the physicians directly explain the patient's condition." (Nurse 2) |
| Setting and sharing goals | Setting and sharing goals | Setting and sharing goals | | "We present each patient with their rehabilitation goals and the expected timeline to reach them. Then, we encourage them by saying, 'Let's work together to reach these goals.'" (PT 3) |
| Support for family and patient interactions | Support for family and patient interactions | | | "For patients who are worried about their families, we arrange opportunities for their families to visit and meet them." (Physiatrist 1) |
| | | Sharing information with family members | | "We involve the family in rehabilitation sessions and teach them how to support the patient, such as assisting with movement. This helps deepen the family's understanding of the patient's condition." (PT 1) |
| Providing practice in increasing the independent level of activities in the hospital | | Feedback about improvement in activities of daily living* | | "When a patient is able to do something independently, like go to the bathroom, we praise their progress and give them positive feedback." (PT 3) |
| Motivating patients by providing feedback on improvements | | Feedback using numerical data | | "We use numerical data to show patients how they have improved and provide feedback. Quantitative data helps patients understand their progress." (OT 1) |
| | | Feedback using videos | | "We record patients' movements on video and show them the video to explain what has improved and what we will be working on next." (PT 2) |
| | | Feedback about improvement in activities of daily living* | | "We use measurable outcomes, like the 10-meter walk test, to show how much their walking has improved in daily life." (PT 3) |
| Providing information on return to work and support for life after discharge | | Assistance in coordinating return to work | | "When a patient is planning to return to work, we simulate their job tasks to assess and train them. We also help coordinate with their workplace." (SLP 4) |
| | | | Providing information for outpatient visits | "We explain that they can continue improving after discharge by using long-term care services or attending outpatient rehabilitation." (Physiatrist 2) |
| | | | Introducing procedures for exercises and assistance methods for the home | "We recreate their home environment as closely as possible and have them practice tasks. If something is difficult, we consider using assistive devices." (OT 5) |
| Providing information about care services | | | Providing information about care services | "We ensure that patients have access to the necessary information before discharge. Since many patients don't know much about the long-term care system, we encourage them to consult with a specialist." (Care Worker 1) |
| Practicing movements needed for life after discharge | | | Simulating of motion required after discharge | "We simulate boarding and exiting buses or trains, and have patients practice those tasks in a recreated environment." (OT 5) |
| | | | Home visit and investigation | "We visit patients' houses to identify barriers. Based on that information, we either train them in the hospital or suggest home modifications." (OT 4) |
| | | | Guidance for self-exercises | "We suggest home exercises that the patient can safely and easily do after discharge." (SLP 4) |

* Factors associated with multiple factors.

Table 4. Correlations between patients' anxiety and the types of interventions in the entire duration of hospitalization.

| | | Types of interventions | | | | | | | |
|---|---|---|---|---|---|---|---|---|---|
| | | Clarifying the rehabilitation content and plan | Setting and sharing goals | Support for family and patient interactions | Providing practice in increasing the independent level of activities in the hospital | Motivating patients by providing feedback on improvements | Providing information on return to work and support for life after discharge | Providing information about care services | Practicing movements needed for life after discharge |
| Patients' anxieties | Prospects for rehabilitation plans | ✓ | ✓ | | | ✓ | | | |
| | Prognosis of physical function | ✓ | ✓ | | | ✓ | | | |
| | Family situation | | | ✓ | | | | | |
| | Hospital life | ✓ | | | ✓ | | | | |
| | Prospects of social life | ✓ | ✓ | ✓ | | | ✓ | ✓ | ✓ |
| | Life at home after discharge | | | | | | ✓ | ✓ | ✓ |

patients' physical condition after disease onset, the unfamiliar environment, and the family situation seemed to form the main contents of anxieties. Medical staff appear to respond to these anxieties by explaining rehabilitation treatment plans, setting and sharing goals, and facilitating support for family and patient interactions. In the middle phase, the patients may gradually get used to the hospital environment and the contents of anxieties may change, considering the prospects and progress of rehabilitation treatments. During this phase, in addition to the interventions used in the early phase, medical staff use various interventions to provide feedback on the progress of rehabilitation treatments. In the late phase, the patients' anxieties are mainly related to life at home after discharge. Interventions also change during this phase to those providing information about nursing care services and outpatient visits.

Previous studies have reported that patients in the acute phase tend to have anxieties about surgical treatments [41,42] and the hospital environment [43,44], whereas those in the chronic phase tend to have anxieties about the possibility of disease recurrence [45]. In this study targeting patients in the convalescent rehabilitation ward, anxieties about the hospital environment, but not about surgical treatment and the possibility of disease recurrence, were also identified. This discrepancy may be attributed to the differences in patient characteristics between the present and previous studies. The target participants in our study included patients in the subacute phase admitted to the convalescent rehabilitation ward. They already received acute medical care for their disease and their physical condition was relatively stable. Therefore, anxieties such as surgical treatment and the possibility of disease recurrence may not be identified. Furthermore, this study demonstrated that the patients in the convalescent rehabilitation ward have anxieties about anticipating an improvement in their physical and cognitive functions and life after discharge, which are elements addressed by rehabilitation treatments.

As for interventions to address these anxieties, our study showed that medical staff provide patient education and interactions. Specifically, they tend to provide explanations of rehabilitation plans, set and share rehabilitation goals, and provide feedback on rehabilitation progress through communication with patients. These results are in line with previous

studies demonstrating that non-psychologist medical staff implement patient education and interactions with medical staff as interventions in both acute [23,46] and chronic phases [23,46,47]. It has been suggested that these interventions, especially goal setting and feedback, not only ensure that patients are in better psychological states but also contribute to an improvement in the physical activity level and functions of patients undergoing rehabilitation [48,49]. Therefore, a part of the interventions to address patients' anxieties identified in this study may help patients be in better psychological states and also improve their rehabilitation outcomes.

An important finding of this study is that patients' anxieties may change depending on the hospitalization phases. Although patients' anxieties about the prognosis of physical function and the prospects of social life were found throughout the hospitalization phases, when focusing on each phase, anxieties about the prospects for the rehabilitation plans and the unfamiliar environment emerged in the early to middle phases of hospitalization. These anxieties were not identified in the late phase of hospitalization, but those about life after discharge were identified. One of the possibilities that lead to these changes may involve the patients' characteristics in the convalescent rehabilitation ward. The patients in the sub-acute phase can better improve their physical functions and abilities [30,31]. Patients in the early phase of hospitalization may expect intensive rehabilitation treatments, but may also have anxieties about uncertainty regarding future improvements in their physical functions and abilities. In the late phase of hospitalization when the time of discharge is decided, patients seem to anticipate their own physical functions at the time of discharge; therefore, they may not have anxiety about future rehabilitation plans for the rest of the hospital stay. Instead, the main anxieties seem to be about the life after discharge, which is unpredictable for the patients. Specifically, the environment after discharge differs from the hospital, potentially reducing chances for the patients to contact medical staff. Therefore, even if patients want to know their own current state from the medical perspective and rehabilitation plans, it may be difficult for them to receive information as they could in the hospital due to reduced chances of contact with medical staff after discharge. Additionally, the changes in the environment from the hospital to that after discharge could leave patients feeling anxious about their ability to perform the movements they learned during rehabilitation in the hospital.

Another possibility that contributes to changes in patients' anxieties across phases may be the characteristics of the convalescent rehabilitation wards. In these wards, the rehabilitation treatments focus on both the improvement of physical functions and patients' return to social life, such as living at home and returning to work [30], but they are not implemented at the same phase. In general, the interventions for physical functions are implemented in the early to middle phases of hospitalization, and the focus of interventions then shifts to preparation for returning home and to social life. This shift appears to be similar to the changes in the content of patients' anxieties across phases of hospitalization identified in this study. Although we assumed that interventions would change according to changes in patients' anxieties, it is also plausible that phase-dependent changes in the focus of interventions may lead to changes in the anxieties. This remaining and intriguing question of causality between the content of patients' anxieties and interventions should be addressed in future studies.

Regarding the clinical implications for healthcare professionals, the present findings suggest the possibility of developing and utilizing tools or checklists in clinical practice that summarize the contents of psychological anxieties and corresponding interventions according to hospitalization phases. Such tools may enable even less clinically experienced staff to provide effective and timely interventions. In addition, for department managers (i.e., public managers), educational efforts can be made to enhance the understanding among medical staff that psychological anxieties in patients are phase-specific, thereby promoting more effective psychosocial intervention during hospitalization. Furthermore, psychological distress has been shown to vary not only over several weeks or months, as observed in the present study, but also within a single day [50], thus necessitating timely interventions that correspond to these changes. Appropriate psychosocial interventions may help prevent individuals from depression and anxiety disorders, which in turn could reduce overall healthcare costs at the societal level. From a policymaker's perspective, the present findings may offer valuable insights into the development of structured frameworks for psychosocial care, specifying when and how such care should be provided during different phases of hospitalization.

## Study limitations

This study has a limitation, and caution should be exercised in generalizing the findings. Although the study provides an overview of the contents of patients' anxieties and interventions based on the medical staff's perspectives, we did not directly investigate the patients' perspectives. As there may be differences in the factors that patients and medical staff perceive as influencing patients' psychological well-being [51], the present findings cannot rule out the possibility of a gap between their perspectives. Further research is required to verify whether the findings of this study are also valid from the patients' perspectives. As explained above, however, patients seem to respond based on their own diseases and individual experiences with hospitalization, resulting in biased responses. Therefore, further research needs to recruit a larger number of patients to minimize the effects of such individual biases. In addition, all participants in this study were recruited from a single convalescent rehabilitation ward in one country, which could restrict the applicability of our findings to other institutions or cultural contexts. In this study, we determined that theoretical saturation had been achieved when no new themes emerged during three consecutive interviews, regardless of the participant's professional role. Although the major clusters derived from the text-mining analysis appeared consistent across professions, the question of whether there are profession-specific differences in perception and expression remains intriguing and is worthy of future study.

## Conclusions

In conclusion, despite the study's limitations, we revealed that the contents of patients' anxieties and the interventions of medical staff to address these anxieties change depending on the hospitalization phases. These findings provide useful information regarding the appropriate ways and timing of involvement to keep patients in better psychological states.

## Supporting information

**S1 Fig. Cluster dendrogram of the patients' anxieties in the early phase of hospitalization.** Cluster 1, prognosis of physical function; Cluster 2, differences from the acute care ward; Cluster 3, prospects of social life; Cluster 4, prognosis of cognitive disorders; Cluster 5, prospects for rehabilitation plans; and Cluster 6, family situation. Dotted vertical line: Threshold of the agglomeration dissimilarity coefficient.
(DOCX)

**S2 Fig. Cluster dendrogram of the patients' anxieties in the middle phase of hospitalization.** Cluster 1, prognosis of physical function; Cluster 2, prospects of social life; Cluster 3, prospects for rehabilitation plans; and Cluster 4, lack of feelings of improvement. Dotted vertical line: Threshold of the agglomeration dissimilarity coefficient.
(DOCX)

**S3 Fig. Cluster dendrogram of the patients' anxieties in the late phase of hospitalization.** Cluster 1, lack of information about the medical follow-up system after discharge; Cluster 2, decline in physical function after discharge; Cluster 3, differences in the physical and interpersonal environments between the hospital and home; and Cluster 4, role reacquisition (return to work, household chores). Dotted vertical line: Threshold of the agglomeration dissimilarity coefficient.
(DOCX)

**S4 Fig. Cluster dendrogram of the interventions for patients' anxieties in the early phase of hospitalization.** Cluster 1, explanation of the current situation by medical staff; Cluster 2, setting and sharing goals; Cluster 3, support for family and patient interactions; and Cluster 4, explanation of the rehabilitation treatment plan by physiatrists. Dotted vertical line: Threshold of the agglomeration dissimilarity coefficient.
(DOCX)

**S5 Fig. Cluster dendrogram of the interventions for patients' anxieties in the middle phase of hospitalization.**
Cluster 1, sharing information with family members; Cluster 2, setting and sharing goals; Cluster 3, assistance in coordinating return to work; Cluster 4, feedback using numerical data; Cluster 5, feedback about improvement in activities of daily living; Cluster 6, feedback using videos; and Cluster 7, explanation of the rehabilitation treatment plan by a physiatrist. Dotted vertical line: Threshold of the agglomeration dissimilarity coefficient.
(DOCX)

**S6 Fig. Cluster dendrogram of interventions for patients' anxieties in the late phase of hospitalization.** Cluster 1, simulation of movements required after discharge; Cluster 2, providing information about care services; Cluster 3, providing information for outpatient visits; Cluster 4, guidance for self-exercises; Cluster 5, home visit and investigation; and Cluster 6, guidance of movements procedures and assistance methods. Dotted vertical line: Threshold of the agglomeration dissimilarity coefficient.
(DOCX)

**S1 Table. Correlations between patients' anxiety and the types of interventions in the early phase of hospitalization.**
(DOCX)

**S2 Table. Correlations between patients' anxiety and the types of interventions in the middle phase of hospitalization.**
(DOCX)

**S3 Table. Correlations between patients' anxiety and the types of interventions in the late phase of hospitalization.**
(DOCX)

**S1 File. Interview topic guide and prompts.**
(DOCX)

**S2 File. Interview Data (Japanese).**
(XLSX)

## Acknowledgments

We thank all the participants from Fujita Health University Hospital and Editage (www.editage.jp) for English language editing.

## Author contributions

**Conceptualization:** Taiki Yoshida, Yoshitaka Wada, Yohei Otaka.

**Data curation:** Taiki Yoshida, Yoshitaka Wada, Shintaro Uehara, Yohei Otaka.

**Formal analysis:** Taiki Yoshida, Yoshitaka Wada, Shintaro Uehara, Kazuki Ushizawa, Yohei Otaka.

**Funding acquisition:** Taiki Yoshida.

**Investigation:** Taiki Yoshida, Yoshitaka Wada, Asuka Hirano.

**Methodology:** Taiki Yoshida.

**Project administration:** Taiki Yoshida, Yohei Otaka.

**Resources:** Taiki Yoshida.

**Supervision:** Shintaro Uehara, Yohei Otaka.

**Validation:** Taiki Yoshida, Yohei Otaka.

**Visualization:** Taiki Yoshida.

**Writing – original draft:** Taiki Yoshida, Yoshitaka Wada, Shintaro Uehara, Asuka Hirano, Kazuki Ushizawa, Hirofumi Maeda, Daisuke Matsuura, Yohei Otaka.

**Writing – review & editing:** Taiki Yoshida, Yoshitaka Wada, Shintaro Uehara, Asuka Hirano, Kazuki Ushizawa, Hirofumi Maeda, Daisuke Matsuura, Yohei Otaka.

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
