## [Decision Letter · Decision Letter 0]

24 Apr 2025

Dear Dr. Otaka,

Thank you for submitting your manuscript to PLOS ONE. After careful consideration, we feel that it has merit but does not fully meet PLOS ONE’s publication criteria as it currently stands. Therefore, we invite you to submit a revised version of the manuscript that addresses the points raised during the review process.

We look forward to receiving your revised manuscript.

Kind regards,

Maheshkumar Baladaniya

Academic Editor

PLOS ONE

Journal Requirements:

https://journals.plos.org/plosone/s/f   ile?id=wjVg/PLOSOne_formatting_sample_main_body.pdf and

2 . Thank you for stating in your Funding Statement:

“This study was supported by JSPS KAKENHI Grant-in-Aid for Early-Career Scientists (Grant no. 22K17579).”

4. We note that your Data Availability Statement is currently as follows: [he original contributions presented in the study are included in the article/Supplementary material, further inquiries can be directed to the corresponding author.]

Additional Editor Comments (if provided):

Reviewers has displayed some major flaws that should be reviewed and resolved to meet the acceptance criteria.

Reviewers' comments:

Reviewer's Responses to Questions

**Comments to the Author**

1. Is the manuscript technically sound, and do the data support the conclusions?

Reviewer #1: Yes

Reviewer #2: Yes

2. Has the statistical analysis been performed appropriately and rigorously?

Reviewer #1: Yes

Reviewer #2: Yes

3. Have the authors made all data underlying the findings in their manuscript fully available?

Reviewer #1: Yes

Reviewer #2: Yes

4. Is the manuscript presented in an intelligible fashion and written in standard English?

Reviewer #1: Yes

Reviewer #2: Yes

Reviewer #1: This is a topic of great scientific and social relevance, explored through a well-developed study that deserves recognition for its qualitative approach. In a field of research where quantitative studies predominate, it is particularly valuable to find studies that seek to deeply understand the complexity of human phenomena from the perspective of the individuals involved. While qualitative methods do not allow for the establishment of cause-and-effect relationships or statistical generalizations, their contribution lies in the richness of narratives, the contextualized understanding of experiences, and the ability to reveal nuances that would hardly emerge from purely quantitative approaches.

In this sense, the study already possesses intrinsic merit by proposing a well-structured qualitative methodology, with clear graphical presentation and cohesive writing. However, there is room for significant improvement that could further enhance the quality and impact of the research.

The introduction, for instance, could be enriched through a more comprehensive and updated literature review that not only contextualizes the investigated issue but also connects it to similar studies conducted in both local and international contexts. Such a review would anchor the study within a broader research panorama, reinforcing its relevance and originality. It would be especially beneficial to include tested intervention models, preventive strategies, and effective care actions in order to demonstrate how the literature has addressed the situations explored in this study and to what extent the current research contributes to this collective effort to improve health practices.

Additionally, the text would benefit from a more gradual and fluid introduction. The opening paragraph should primarily serve to situate the reader within the research problem, explaining its social and scientific significance before delving into theoretical definitions and conceptual frameworks. This would make the text more accessible and cohesive, facilitating comprehension of the topic from the outset.

From a methodological perspective, the study presents a solid structure but still lacks clarification on essential aspects. One such aspect is the sample definition. The number of participants — 17 physicians — needs to be justified. It is necessary to explain why this specific number was chosen and what inclusion and exclusion criteria were adopted. Comparing this choice with similar studies may help to support the decision methodologically and strengthen the study’s credibility.

Another crucial point concerns the data collection and analysis process. Qualitative methodology demands transparency and a rigorous level of detail to ensure the reliability of results. Therefore, it is recommended that the study present all guiding questions used in the interviews, as well as the interview protocol. It is also essential to describe, in as much detail as possible, the participant selection process, the conditions under which the interviews took place, and whether there were multiple rounds of interviews with the same participants. Such information enables a better understanding of the data production context and allows for the assessment of quality, while also helping to minimize bias and subjectivity.

Likewise, the inclusion of excerpts from participants’ responses is highly recommended. Beyond enriching the study’s narrative, these excerpts lend authenticity to the analysis and highlight the rigorous treatment of the data. Direct quotations, properly contextualized, reinforce the credibility of the findings and provide a deeper insight into the experiences reported.

Furthermore, the role of the researchers themselves should be made explicit. In qualitative research, it is crucial to acknowledge that the researcher is also part of the research context, particularly when conducting interviews or participating in data collection and analysis. Recognizing this dynamic strengthens the transparency of the research and adds legitimacy to the findings.

The discussion of results is one of the most central elements of the study and should be addressed with appropriate depth. It is vital that the findings are critically linked to the existing literature, highlighting similarities, divergences, and any novel contributions. This critical analysis not only demonstrates mastery of the subject area but also reinforces the relevance of the findings for scientific advancement. Moreover, the inclusion of practical and applicable suggestions aimed at healthcare professionals, public managers, and policymakers would be pertinent.

Another element that could further reinforce the study’s relevance is the proposal of future research directions. Identifying gaps in the current knowledge and suggesting promising paths for new studies reflects a strong commitment to the continuity of scientific inquiry and expands the reach of the research.

In summary, this is a study with great potential, supported by a solid methodological foundation and focused on a highly relevant topic. With some improvements — particularly in terms of theoretical grounding, methodological transparency, and analytical depth — the work could achieve even greater impact.

Reviewer #2: General Comments

The manuscript is methodologically sound, clearly structured, and provides useful insights into how rehabilitation staff perceive and address the emotional and psychological challenges of patients. However, a few areas require clarification, and some improvements could strengthen the manuscript’s rigor and practical relevance.

Major Comments

Abstract

Well-structured, but it would benefit from reporting specific examples of anxieties and interventions to provide more concrete insight.

Introduction

Consider adding a brief literature comparison with other qualitative studies addressing similar topics in different cultural or clinical settings.

Methods

Clarify if interview transcripts were returned to participants for review (member checking).

While the rationale for not interviewing patients (concerns about recall bias and individual variability) is mentioned, it still raises a limitation in terms of representativeness of patient voices. Include a more detailed justification for the exclusive use of medical staff input and discuss whether a triangulation approach involving both patient and staff perspectives might offer a richer analysis in future work.

Results

The current clustering provides breadth but lacks deep thematic interpretation. Complement the cluster results with qualitative excerpts or mini-case examples from interviews to give depth and voice to the findings.

Discussion

The implications for practice could be expanded. For example:

How can staff training be improved using these insights? Could an intervention toolkit or checklist be developed from these findings?

Author could write in implication section below phrase to strengthen the study findings.

“Recent evidence has demonstrated that psychological distress and suicide attempts follow distinct temporal patterns, with increased risk during afternoon and evening hours, particularly among vulnerable populations such as young and married individuals [Ref:sheikh et al. doi; https://doi.org/10.1016/j.bbii.2024.100072]. Our findings highlight the need for timely psychosocial interventions that are sensitive to daily fluctuations in mood and anxiety, especially in rehabilitation settings where patients face uncertainties about recovery and discharge.”

The study involved 17 staff members from a single hospital, potentially limiting generalizability.

Acknowledge this more clearly as a limitation and discuss if saturation was reached in the interviews. Consider explaining if there was any variation in responses between different professional roles (OT, PT, nurse, etc.).

**Do you want your identity to be public for this peer review?** For information about this choice, including consent withdrawal, please see our Privacy Policy

Reviewer #1: No

Reviewer #2: No

---

## [Author Response · Author response to Decision Letter 1]

9 Jun 2025

Point-by-Point Responses to Reviewers’ Comments

Reviewer 1

Comment 1:

The introduction, for instance, could be enriched through a more comprehensive and updated literature review that not only contextualizes the investigated issue but also connects it to similar studies conducted in both local and international contexts. Such a review would anchor the study within a broader research panorama, reinforcing its relevance and originality. It would be especially beneficial to include tested intervention models, preventive strategies, and effective care actions in order to demonstrate how the literature has addressed the situations explored in this study and to what extent the current research contributes to this collective effort to improve health practices.

Additionally, the text would benefit from a more gradual and fluid introduction. The opening paragraph should primarily serve to situate the reader within the research problem, explaining its social and scientific significance before delving into theoretical definitions and conceptual frameworks. This would make the text more accessible and cohesive, facilitating comprehension of the topic from the outset.

Response:

Thank you for your valuable suggestion. In response to your comment and Reviewer 2’s Comment 2, we have revised the introduction to provide a more comprehensive and updated literature review. Specifically, we additionally introduced structured interventions (a goal-setting intervention, person-centered care strategies, etc.) aimed to address psychological distress in hospitalized patients. Then, we incorporated this information to reconsider how our research is situated within the existing findings and clarify the purpose and originality. In addition, according to the reviewer’s suggestion, we restructured the opening paragraph in the introduction section to gradually introduce readers to the research problem by first explaining its social and scientific significance, then presenting the theoretical and conceptual frameworks.

We have significantly modified the paragraph in the introduction section from lines 42-44, 46-48, and 54-66 in the revised manuscript.

Comment 2:

From a methodological perspective, the study presents a solid structure but still lacks clarification on essential aspects. One such aspect is the sample definition. The number of participants — 17 physicians — needs to be justified. It is necessary to explain why this specific number was chosen and what inclusion and exclusion criteria were adopted. Comparing this choice with similar studies may help to support the decision methodologically and strengthen the study’s credibility.

Response:

We appreciate the reviewer’s comment and have included a detailed description of the inclusion criteria in the revised manuscript. Regarding the sample size, previous qualitative interview studies suggested that theoretical saturation can be achieved with 12 participants (Guest et al., 2006). Based on this, we aimed to sample beyond this number while monitoring saturation. We achieved theoretical saturation when no new themes emerged from three consecutive interviews and decided to cease recruitment. We have added the above information from lines 152-156 and 159-165 in the revised manuscript.

Comment 3:

Another crucial point concerns the data collection and analysis process. Qualitative methodology demands transparency and a rigorous level of detail to ensure the reliability of results. Therefore, it is recommended that the study present all guiding questions used in the interviews, as well as the interview protocol.

It is also essential to describe, in as much detail as possible, the participant selection process, the conditions under which the interviews took place, and whether there were multiple rounds of interviews with the same participants. Such information enables a better understanding of the data production context and allows for the assessment of quality, while also helping to minimize bias and subjectivity.

Response:

Thank you for your valuable comment. As described in the Methods section, we asked participants the following two guiding questions during the interviews: “What are the anxieties in patients during the early / middle / late phase of hospitalization in a convalescent rehabilitation ward?” and “How do you address those anxieties?” In addition to these main questions, we used a set of supplementary prompts to facilitate the interviews. We have now added the complete list of guiding questions and prompts (original Japanese version and informal English translation version) have been added as supplementary data.

Regarding the participant selection process, we displayed posters describing the purpose and outline of the study in the hospital wards where potential participants worked. Individuals who expressed their willingness to participate were recruited for the study. Each participant was interviewed only once. We have revised the manuscript to include these methodological details (from lines 131-136, 152-156).

Comment 4:

Likewise, the inclusion of excerpts from participants’ responses is highly recommended. Beyond enriching the study’s narrative, these excerpts lend authenticity to the analysis and highlight the rigorous treatment of the data. Direct quotations, properly contextualized, reinforce the credibility of the findings and provide a deeper insight into the experiences reported.

Response:

Thank you for your insightful suggestion. To enhance the credibility of the analysis and provide a deeper insight into the experiences reported, we have added representative quotations from participants’ responses to Tables 2 and 3.

Comment 5:

Furthermore, the role of the researchers themselves should be made explicit. In qualitative research, it is crucial to acknowledge that the researcher is also part of the research context, particularly when conducting interviews or participating in data collection and analysis. Recognizing this dynamic strengthens the transparency of the research and adds legitimacy to the findings.

Response:

In response to this comment, we have revised the manuscript to explicitly describe the roles and backgrounds of the researchers involved in data collection and analysis. This revision aligns with Domain 1 of the COREQ (Consolidated Criteria for Reporting Qualitative Research) guidelines. Specifically, we clarified the relationship between the interviewers and participants, each researcher’s clinical and research experience, and the method’s procedures.

We have added the above information from lines 131-136, 140-144, and 183-185, in the revised manuscript.

Comment 6:

The discussion of results is one of the most central elements of the study and should be addressed with appropriate depth. It is vital that the findings are critically linked to the existing literature, highlighting similarities, divergences, and any novel contributions. This critical analysis not only demonstrates mastery of the subject area but also reinforces the relevance of the findings for scientific advancement. Moreover, the inclusion of practical and applicable suggestions aimed at healthcare professionals, public managers, and policymakers would be pertinent.

Response:

In response, we revised the Discussion section to offer a more critical analysis of our results in relation to previous studies. We explained how the interventions we observed align with structured models such as goal-setting strategies and person-centered care. These models have been shown to enhance psychological well-being and functional outcomes in different clinical populations (Kalra et al., 2000; Gustafsson et al., 2015). We included these references to demonstrate the relevance and consistency of our findings within the broader research context.

Furthermore, we provided practical suggestions for how healthcare professionals, public managers, and policymakers may apply our results. Specifically, we emphasized the importance of phase-specific interventions, interdisciplinary communication, and organizational support structures that enable non-psychologist medical staff to deliver timely psychosocial care in rehabilitation settings.

We have added the above information from lines 318-331, in the revised manuscript.

Comment 7:

Another element that could further reinforce the study’s relevance is the proposal of future research directions. Identifying gaps in the current knowledge and suggesting promising paths for new studies reflects a strong commitment to the continuity of scientific inquiry and expands the reach of the research.

Response:

We appreciate the reviewer’s suggestion. As mentioned in line 289 in the previous manuscript, we discussed future research directions. However, we have now added detailed descriptions to identify gaps in the current knowledge and suggest promising paths for new research. Specifically, drawing on previous studies, we highlighted that patients and medical staff may perceive different factors as influencing psychological aspects, thereby emphasizing the importance of incorporating both perspectives.

We have added the above information from lines 335-338 in the revised manuscript.

Reviewer 2

Comment 1:

Abstract

Well-structured, but it would benefit from reporting specific examples of anxieties and interventions to provide more concrete insight.

Response:

Thank you for your helpful comment. We agree that some descriptions of anxiety, including “hospital life,” “family situation,” and “prospects of social life,” lacked sufficient clarity and detail. To address this, we added specific examples to the Results section of the abstract to make these points more concrete and easier to understand. To stay within the journal’s 300-word limit, we revised and condensed the other sections of the abstract.

Comment 2:

Introduction

Consider adding a brief literature comparison with other qualitative studies addressing similar topics in different cultural or clinical settings.

Response:

Thank you for your valuable suggestion. As response to Reviewer 1’s Comment #1, we revised the introduction to provide a more comprehensive and updated literature review. We added comparisons with related qualitative studies conducted in different clinical and cultural settings (please see the response to Reviewer 1’s Comment #1 for more detail).

Comment 3:

Methods

Clarify if interview transcripts were returned to participants for review (member checking).

While the rationale for not interviewing patients (concerns about recall bias and individual variability) is mentioned, it still raises a limitation in terms of representativeness of patient voices. Include a more detailed justification for the exclusive use of medical staff input and discuss whether a triangulation approach involving both patient and staff perspectives might offer a richer analysis in future work.

Response:

Thank you for this important comment. In this study, we did not return interview transcripts or cluster results to participants for review, a process known as “member checking.” However, we took careful steps to ensure the accuracy and integrity of the data. First, a professional transcription service independent of the research team transcribed the interviews, minimizing the possibility of researcher bias. Second, multiple researchers (TY, AH, and KU) thoroughly discussed the naming and interpretation of clusters derived from the text mining and hierarchical cluster analysis (HCA) results. These researchers reviewed both the keywords and original narratives to ensure that the cluster labels accurately reflected the participants’ intentions.

Regarding the decision to interview only medical staff, we acknowledge the limitation of not directly investigating patients’ perspectives. As stated in the manuscript, this decision was based on the potential for recall bias and variability in subjective experiences among patients. We agree that triangulation can enhance the validity and richness of the analysis by incorporating the perspectives of both staff and patients. In particular, patients who have experienced both the acute and recovery phases may offer unique insights. We have included this as a direction for future research.

We have added the above information from lines 95-96, 167-170, 187-188, and 342-348 in the revised manuscript.

Comment 4:

Results

The current clustering provides breadth but lacks deep thematic interpretation. Complement the cluster results with qualitative excerpts or mini-case examples from interviews to give depth and voice to the findings.

Response:

Thank you very much for this insightful suggestion. In response, we added direct excerpts from the participants’ interview responses to Tables 2 and 3 of the revised manuscript, one for each cluster. This was done to enhance thematic interpretation and provide deeper insight into the findings. We believe that including these quotes gives readers a better understanding of the meaning and context of each cluster. It also gives voice to the participants and strengthens the narrative depth of the results.

Comment 5:

The implications for practice could be expanded. For example: How can staff training be improved using these insights? Could an intervention toolkit or checklist be developed from these findings? Author could write in implication section below phrase to strengthen the study findings.“Recent evidence has demonstrated that psychological distress and suicide attempts follow distinct temporal patterns, with increased risk during afternoon and evening hours, particularly among vulnerable populations such as young and married individuals [Ref:sheikh et al. doi; https://doi.org/10.1016/j.bbii.2024.100072]. Our findings highlight the need for timely psychosocial interventions that are sensitive to daily fluctuations in mood and anxiety, especially in rehabilitation settings where patients face uncertainties about recovery and discharge.”

Response:

Thank you for this constructive and insightful comment. Based on your suggestion, we have expanded the clinical implications in the revised manuscript. We now describe how the study findings may be applied to improve staff training and support the development of a practical toolkit or checklist. In particular, we highlight that the identified temporal changes in patients’ anxieties and the corresponding interventions can guide less experienced rehabilitation professionals in providing timely and phase-appropriate psychosocial care. Moreover, in line with the recent findings (Sheikh et al., 2024), which demonstrated temporal patterns in psychological distress and suicide risk, we emphasize the importance of time-sensitive psychosocial interventions, especially in rehabilitation settings where patients face uncertainty about recovery and discharge.

We have added the above information from lines 318-331 in the revised manuscript.

Comment 6:

The study involved 17 staff members from a single hospital, potentially limiting generalizability.

Acknowledge this more clearly as a limitation and discuss if saturation was reached in the interviews. Consider explaining if there was any variation in responses between different professional roles (OT, PT, nurse, etc.).

Response:

We appreciate the reviewer’s insightful comment. In the Limitations section, we have acknowledged that the number of participants was limited and that all were recruited from a single convalescent rehabilitation hospital, which may affect the generalizability of our findings.

To confirm theoretical saturation, we monitored the emergence of new themes and determined that saturation had been reached when no new themes emerged in the interviews of three consecutive participants, regardless of their professional role.

While the major clusters derived from the text-mining analysis were consistent across professional roles, we cannot rule out the possibility of subtle narrative differences depending on professional roles. Although the current sample size is insufficient for a detailed analysis, this is an interesting topic for future research.

We have added the above information from lines 159-165 and 342-348, in the revised manuscript.

---

## [Decision Letter · Decision Letter 1]

17 Jul 2025

Medical staff’s perspectives on patients’ anxieties and interventions in a rehabilitation ward: A qualitative study

PONE-D-25-07050R1

Dear Dr. Otaka,

We’re pleased to inform you that your manuscript has been judged scientifically suitable for publication and will be formally accepted for publication once it meets all outstanding technical requirements.

Kind regards,

Maheshkumar Baladaniya

Academic Editor

PLOS ONE

Additional Editor Comments (optional):

Author has made the changes. Manuscript is acceptable for the publication.

Reviewers' comments:

Reviewer's Responses to Questions

**Comments to the Author**

Reviewer #2: All comments have been addressed

Reviewer #3: All comments have been addressed

2. Is the manuscript technically sound, and do the data support the conclusions?

Reviewer #2: Yes

Reviewer #3: Yes

3. Has the statistical analysis been performed appropriately and rigorously?

Reviewer #2: Yes

Reviewer #3: Yes

4. Have the authors made all data underlying the findings in their manuscript fully available?

Reviewer #2: Yes

Reviewer #3: Yes

5. Is the manuscript presented in an intelligible fashion and written in standard English?

Reviewer #2: Yes

Reviewer #3: Yes

Reviewer #2: Authors have addressed all comments, and it is ready for acceptance. However, it depends upon editorial decision.

Reviewer #3: The writers have strengthened the introduction, methodology, and discussion by carefully addressing the prior reviewer's suggestions. Participant quotes and more obvious practical applications have increased the manuscript's impact and clarity. In its current form, the paper is appropriate for publishing, methodologically solid, and well-written.

**Do you want your identity to be public for this peer review?** For information about this choice, including consent withdrawal, please see our Privacy Policy

Reviewer #2: No

Reviewer #3: No

---

## [Editor Report · Acceptance letter]

PONE-D-25-07050R1

PLOS ONE

Dear Dr. Otaka,

I'm pleased to inform you that your manuscript has been deemed suitable for publication in PLOS ONE. Congratulations! Your manuscript is now being handed over to our production team.

Kind regards,

on behalf of

Dr. Maheshkumar Baladaniya

Academic Editor

PLOS ONE